# You Can Learn Tokenization End-to-End with Reinforcement Learning

Sam Dauncey [* 1]    Roger Wattenhofer [1]

## Abstract

Tokenization is a hardcoded compression step which remains in the training pipeline of Large Language Models (LLMs), despite a general trend towards architectures becoming increasingly end-to-end. Prior work has shown promising results at scale in bringing this compression step inside the LLMs' architecture with heuristics to draw token boundaries, and also attempts to learn these token boundaries with straight-through estimates, which treat the problem of drawing discrete token boundaries as a continuous one. We show that these token boundaries can instead be learned using score function estimates, which have tighter theoretical guarantees due to directly optimizing the problem of drawing discrete token boundaries to minimize loss. We observe that techniques from reinforcement learning, such as time discounting, are necessary to reduce the variance of this score function sufficiently to make it practicable. We demonstrate that the resultant method outperforms prior proposed straight-through estimates, both qualitatively and quantitatively at the 100 million parameter scale.

## 1. Introduction

Tokenization is a crucial pre-processing step in the training and inference pipelines of modern LLMs. Standard practice compresses text into symbols representing commonly occurring substrings. This is typically done using algorithms such as Byte-Pair Encoding (BPE), which recursively groups frequently co-occurring byte sequences into individual tokens. State-of-the-art open-source models further augment BPE with numerous hand-crafted decisions. For example, the Gemma-series tokenizers (Rivière et al., 2024) explicitly split digits and preserve whitespace, reflecting the extent to which tokenizer design remains largely artisanal.

A growing line of work seeks to eliminate this BPE step entirely by operating directly on UTF-8 bytes (Xue et al., 2022; Wang et al., 2024; Zheng et al., 2025). This can be viewed as using a tokenizer with a maximally small vocabulary and an effective downsampling rate of 1 token per byte. Recent scaling analyses suggest that downstream loss is optimized by increasing vocabulary size, and thus downsampling rate, with model scale (Tao et al., 2024).

We thus focus our investigation onto methods which admit a growing downsampling rate. Interestingly, BPE itself experiences harsh diminishing returns in downsampling rate as vocabulary size scales. Further, a large vocabulary size can have undesirable downstream effects, such as the emergence of very rare "glitch tokens" (Rumbelow & Watkins, 2023). Marginally increasing the achieved downsampling rate to vocabulary size tradeoff with curricula (Liu et al., 2025), has thus proved fruitful in improving LLM performance.

An alternative family of approaches processes text at the byte level for several transformer layers before downsampling into a shorter sequence of latent tokens (Nawrot et al., 2022; Yu et al., 2023). Some of these methods facilitate a simple modification of the downsampling rate as a hyperparameter. When token boundaries are chosen heuristically—e.g., using whitespace (Slagle, 2024) or spikes in next-byte entropy (Nawrot et al., 2023)—these models have been reported to achieve superior performance to pure BPE transformer models at large scales (Pagnoni et al., 2024).

However, such heuristics raise a natural question: can we improve on these boundary rules by learning tokenization itself from end-to-end training? Prior approaches to this question have focused on straight-through estimators (Nawrot et al., 2023), which craft rules for backpropagating gradients from representations internal to the model. We instead investigate score function estimators, which directly approximate the gradient of the expected loss with respect to the token boundaries. Score function estimators have stronger theoretical guarantees, at the cost of a higher variance.

Our main contributions are as follows:

- We show that we can learn tokenization strategies that align closely with semantic boundaries without any explicit prior structure or inductive bias with a score-function estimator, equipped with variance-reduction

[1]Department of Electrical Engineering, ETH Zürich, Zürich, Switzerland. Correspondence to: Sam Dauncey <sdauncey@ethz.ch>.

*Proceedings of the 43rd International Conference on Machine Learning*, Seoul, South Korea. PMLR 306, 2026. Copyright 2026 by the author(s).

techniques from reinforcement learning.

- We further find that our method qualitatively and quantitatively outperforms prior approaches using straight-through estimators.

- We demonstrate robust performance across a range of downsampling rates.

## 2. Theory & Method

In this section, we motivate our method from first principles. We first present some desiderata for an end-to-end tokenisation method §2.1, then define notation for the autoregressive U-net §2.2. We then present score functions as the canonical way to learn tokenization under this framework §2.3 and present our method for making them practicable §2.4 - 2.7

### 2.1. Desiderata for Designing a End-to-End Tokenization Method

For the reasons outlined in §1, we are interested in bringing the tokenization process inside the architecture and training of an LLM. We propose the following desiderata for such a method to be practicable and general:

- **End-to-end tokenizer training**: the token boundary decisions should be learned to minimise loss, in favor of hand-crafted methods and heuristics.

- **End-to-end architecture**: learned representations at the byte level should be re-used at the token level.

- **Efficiency**: use less than 0.1% additional pretraining compute, on existing hardware, than byte-pair-encoding guided tokenization in the forward & backward passes.

### 2.2. Autoregressive U-Net Architecture and Setup

In this section, we define notation for the autoregressive U-net architecture (Nawrot et al., 2022), depicted in Figure 1. To satisfy our **End-to-end architecture** desideratum and reuse byte-level representations at the token level, the model moves from the byte level to the token level and back, via a single downsampling and a single upsampling step. We leave architectures with several such tokenization stages, or several tokenization levels per sequence, to future work.

Because the model is autoregressive, each token can summarize the byte stream only up to a cutoff index, the token boundary. Tokenizing the byte stream thus amounts to choosing where these jumps fall: an inherently discrete decision, independent of how each token pools its bytes.

The forward pass is then structured as follows:

Let $x_1 \ldots x_N$ be a sequence of input bytes and $d_{enc}, d_{mid}, d_{dec}$ be hyperparameters for the model dimensions.

1. Autoregressively encode the input bytes into byte-level representations $X \in \mathbb{R}^{N \times d_{enc}}$:

$$X = \texttt{encode}(x) \tag{1}$$

2. (Potentially stochastically) predict token boundaries from the byte-level representations, where $a_i = 1$ if we wish to draw a token boundary at $x_i$ and $a_i = 0$ if not.

$$a \sim \pi(X, x) \tag{2}$$

3. Downsample the byte-level representations into token-level representations $X' \in \mathbb{R}^{M \times d_{mid}}$ where $M = \sum_{i=0}^{N} a_i$:

$$X' = \texttt{downsample}(X, a) \tag{3}$$

4. Enrich token-level representations $Y' \in \mathbb{R}^{M \times d_{mid}}$ with an autoregressive feedforward network:

$$Y' = \texttt{mid}(Y) \tag{4}$$

5. Upsample into updated byte-level information $Y \in \mathbb{R}^{N \times d_{dec}}$, potentially carrying encoded byte-level information:

$$Y = \texttt{upsample}(Y', X, a) \tag{5}$$

6. Decode the resulting byte-level representations into predictions of the next bytes $y_i = x_{i+1}$

$$y \sim \texttt{decode}(Y) \tag{6}$$

All the above operations must be autoregressive, for example the $\texttt{decode}$ function must be formulated such that $X'_j$, the downsampled representation at position $j$, depends only on the preceding bytes $X_{\leq i}$, where $i$ is the minimum token index such that $j = \sum_{k=0}^{i} a_k$.

While the score function estimate we detail in the following sections can be flexibly applied to any implementation of the above functions, in contrast to straight-through estimate based approaches which require specialized up/downsamplers. We make the following standard choices for our experiments in §4. $\texttt{mid}$ is a decoder-only transformer with full attention, and $\texttt{encode}, \texttt{decode}$ are decoder-only transformers with sliding window attention and linear embedding/unembedding matrices respectively. Our implementation of $\texttt{downsample}$ simply selects the values of $X'$ which correspond to $a_i = 1$ values:

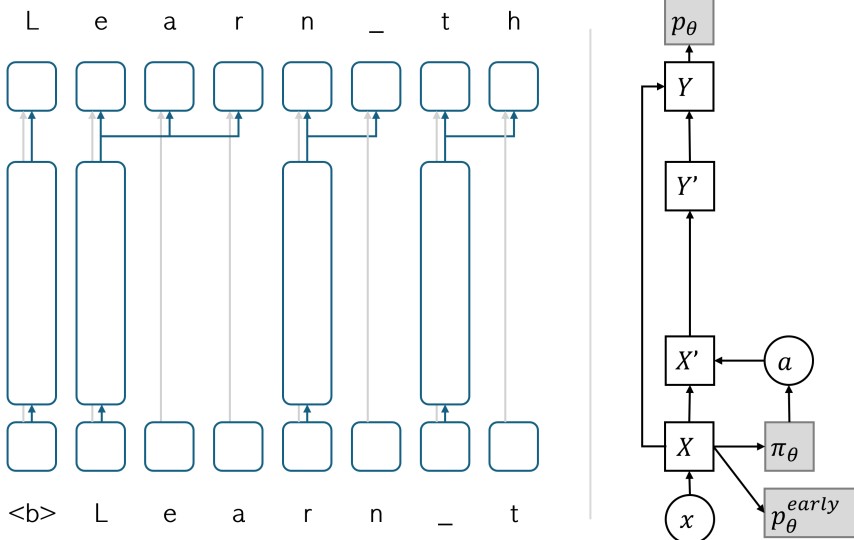

*Figure 1.* [Left] An example of the autoregressive U-net architecture (Nawrot et al., 2022), computed with the values $x_{0:7} = $ `Learn t` and $a_{0:7} = 11001010$. Blocks with rounded edges represent transformer blocks, arrows represent flow of representations. [Right] the stochastic computation graph (Schulman et al., 2015) of a forward pass of our method, deterministic nodes in squares, stochastic nodes in circles, distributions in gray.

$$\texttt{downsample}(X, a)_i = X'_j \quad (7)$$

$$\text{for } j \text{ the minimum value such that } j = \sum_{k=0}^{i} a_k \quad (8)$$

For upsampling, we employ a simple distribute-then-add:

$$\texttt{upsample}(Y', X, a)_j = X_j + Y_i \quad \text{for } i = \sum_{k=0}^{j} a_k \quad (9)$$

### 2.3. Score Function Estimation for Tokenization

As we established in §2.2, tokenizing the byte stream is a discrete choice of where to place token boundaries. The loss is therefore not differentiable with respect to this choice, so to satisfy our **End-to-end tokenizer training** desideratum the model must explore strategies for drawing these token boundaries stochastically.

The model next-token cross-entropy loss outputted is thus conditional on a sampled tokenization strategy $a \sim \pi_\theta$, so we can formulate the problem of simultaneously learning the (potentially shared) parameters of the autoregressive model $p_\theta$ and the gating strategy $\pi_\theta$ as minimizing the next-token cross-entropy marginalized over all $a$:

$$\log p_\theta(y|x) = \mathbb{E}_{a \sim \pi_\theta} \log p_\theta(y|a, x). \quad (10)$$

This is a case of a stochastic computation graph as in Schulman et al. (2015), who show that the gradient of this likelihood can be computed as the expectation of the sum of two separate gradients, one being the standard next-token cross-entropy loss conditioned on the tokenization strategy and a correction term which can be interpreted as applying REINFORCE (Williams, 1992) to the tokenization strategy $\pi_\theta$ with the next-token cross-entropy as the reward:

$$\nabla_\theta \mathbb{E}_{a \sim \pi_\theta} \log p_\theta(y|a, x) \quad (11)$$

$$= \mathbb{E}_{a \sim \pi_\theta} (\underbrace{\nabla_\theta \log p_\theta(y|a, x)}_{\text{conditional loss gradient}} \quad (12)$$

$$+ \underbrace{\log p_\theta(y|a, x) \nabla_\theta \log \pi_\theta(a|x)}_{\text{policy gradient}}). \quad (13)$$

See Appendix A.1 for a complete derivation. This type of estimator is known as a score function estimate. In particular this means that, in the large compute and data limit, performing gradient descent on the policy gradient term above will yield a locally optimal tokenization strategy. No such theoretical guarantee exists for the straight-through estimates we discuss in §3.2.

### 2.4. Reducing the Variance of the Score Function Estimate

In practice, we wish to use a Monte-Carlo estimate of the policy gradient term in equation 11 with a single sample

of $a$ per sequence: using more would break our **Efficiency** desideratum. We empirically find that the naïve REIN-FORCE policy gradient is too noisy to efficiently learn in this setting. In this section, we show how standard techniques from reinforcement learning can be used to de-noise this estimate, solving the corresponding "reward attribution problem": associating which token boundary decisions are responsible for increasing or decreasing the loss in the succeeding tokens.

For this section, we will let $d_{model}$ be the model dimension, `vocab_size` be the number of unique `utf-8` bytes and special characters forming our vocabulary. We will also use the token index of $i$ and the batch index of $b$, which will be omitted where unused.

**Early exit relative rewards**

As our model is autoregressive, we restrict our analysis to the case where the token boundary decision at byte $a_i$ may only depend on the preceding bytes and token boundary decisions, $x_{<i}$ and $a_{<i}$. Thus, the special case of the policy gradient term in eq. (11) treats the token boundary policy $\pi_\theta(a_i|x_{\leq i}, a_{<i})$ as having corresponding rewards $\log p(x_j|a_{<j}, x_{<j})$ for $j > i$.

As $\mathbb{E}_{a \sim \pi_\theta} \nabla_\theta \log \pi_\theta(a_i|x_{\leq i}, a_{<i}) = \mathbf{0}$, we may add any term independent of $a_i$ to these rewards and get a valid policy gradient estimate. The feed-forward nature of the autoregressive U-net architecture presents a natural method to estimate the baseline, tokenization-independent difficulty of predicting the next token by using the early byte-level embeddings to estimate the next-token probability:

$$\log p_\theta^{early}(x_i = t_j|x_{<i}) = \log \texttt{softmax}(W_{early}X_{i-1})_j.$$
$$(14)$$

Where $W_{early} \in \mathbb{R}^{d_{model} \times \texttt{vocab\_size}}$ is an unembedding matrix. This gives a reward with a large part of the tokenization-independent noise subtracted out:

$$R_i = \log p_\theta(x_i|x_{<i}, a_{<i}) - \log p_\theta^{early}(x_i|x_{<i}). \quad (15)$$

We initialize the weights of $W_{early}$ to equal the weights of the final output head to facilitate easy transfer.

**Time discounting**

Summing the above rewards to produce advantages still suffers from too high a variance and due to the reward attribution problem. A common solution to this problem for long-horizon RL is to apply time discounting to the rewards when computing advantages: introducing a small amount of bias into the policy gradient to massively reduce the variance. Intuitively, this decouples the advantages given to far

away parts of the sequence, giving us many approximately independent training stimuli for the token boundaries per sequence.

$$G_i = \sum_{j=0}^{N-i-1} \gamma^j R_{i+j+1} \quad (16)$$

In our experiments, we use a discount factor of $\gamma = 0.99$.

**Batch-relative advantages**

Our $G_i$ values tend to be positive as the final-layer model $p_\theta$ tends to outperform the early-exit model $p_\theta^{early}$. This bias depends on the token index, with the gap being larger for later token indices. To remedy this, we leverage that during batched training we in fact have $B$ such values $G_{i,1} \ldots G_{i,B}$ for a given forward/backward pass, which can be used to center the advantage estimates:

$$A_{i,b} = G_{i,b} - \bar{G}_i \qquad \text{where} \qquad \bar{G}_i = \frac{1}{B}\sum_{b=1}^{B} G_{i,b}. \quad (17)$$

This gives the final policy loss, whose gradient approximates the right hand term of equation (11):

$$\mathcal{L}^\pi = -\sum_{i=0}^{N} \log \pi_\theta(a_i|x_{<i}, a_{<i}) \cdot \texttt{detach}(A_i). \quad (18)$$

Where we use `detach` to emphasize that we do *not* allow gradients to be backpropagated through $A_i$ directly.

## 2.5. Defining the Token Boundary Function

We define a token boundary policy $\pi_\theta$ as a function which gives the probability that our model draws a token boundary at a given index, conditioned on the preceding bytes and token boundaries. We parameterize this probability as the sigmoid of the corresponding logit $l_i$.

$$a_i|x_{\leq i}, a_{<i} \sim \texttt{Bernoulli}(p_i), \quad (19)$$
$$p_i = \pi_\theta(a_i = 1|x_{\leq i}, a_{<i}) = \sigma(l_i). \quad (20)$$

We design the computation of $l_i$ to be of negligible computational cost relative to the total forward pass and sufficiently expressive that our model could learn the token boundary heuristics that have been explored by prior work. Even without explicit training, models already encode rich information about the sequence in their internal representations: for example entropy (Nawrot et al., 2022; Pagnoni et al.,

2024) has been shown to be mediated by directions in the internal representations of models soley trained on next-token prediction (Stolfo et al., 2024). Nonetheless, to express fixed striding strategies, as in MEGABYTE (Yu et al., 2023), we need to give the model access to some sliding window of preceding token boundaries.

Concretely, we compute the raw logit $l_i^{raw}$ by applying a set of linear projections $W_j \in \mathbb{R}^{1 \times d_{model}}$ to the byte-level representation $X_i$ to get a base value for the logit and a series of terms conditional on each token boundary in the window. This operation is amenable to fast computation on modern hardware by pre-computing $W_k X_i$ for all $i, k$ in a single matrix multiply and then performing a fast scan operation.

$$l_i^{raw} = W_0 X_i + \sum_{j=1}^{w} a_{i-j} W_k X_i. \qquad (21)$$

In our experiments, we set a window size of $w = 8$. We additionally study a reduced variant with $w = 1$, in which the boundary logit depends only on the byte representation $X_i$ and the single immediately preceding boundary $a_{i-1}$; we find in §4.2 that this variant performs comparably, indicating that a large window is not required.

With no constraint on the downsample rate, the model will elect to use the computationally expensive strategy of separating every byte with a token boundary. To avoid this, we need to push our model towards a target downsample rate $\bar{\pi}_{target}$, which we discuss further in §2.6. As in attention, we need to scale the raw logits to be approximately uniform at initialization, we further aid stability by adding a $\sigma^{-1}(\bar{\pi}_{target})$ term so that $p_i \approx \bar{\pi}_{target}$ at initialization.

$$l_i^{scaled} = \frac{l_i^{raw}}{D} + \sigma^{-1}(\bar{\pi}_{target}) \qquad (22)$$

In our experiments, we choose a scaling factor of $D = 16$ and a target downsample rate of $\bar{\pi}_{target} = \frac{1}{5}$. To avoid numerical issues caused by exploding logits, we finally apply softcapping (Rivière et al., 2024) to the scaled logits.

$$l_i = \texttt{softcap}(l_i^{scaled}) \qquad (23)$$

During evaluation, we skip this softcapping step, and simply set $l_i = l_i^{scaled}$

### 2.6. Downsample Rate Targeting

We a mechanism to keep the downsample rate close to $\bar{\pi}_{target}$ by applying an even pressure across all the logits $l_{i,b}$ in the batch. We prefer this to operating on the token boundary probabilities, which we find can have unstable results

due to the uneven gradient magnitude of the sigmoid function. Specifically, we apply a negative or positive pressure to the whole batch mean logit $\bar{l} = \frac{1}{NB} \sum_{i,b} l_{i,b}$ if the mean token boundary probability $\bar{p} = \frac{1}{NB} \sum_{i,b} p_{i,b}$ exceeds or falls short of the target downsample rate respectively. We operationalize this with the loss:

$$\mathcal{L}^{target} = \bar{l} \cdot \texttt{detach}(\bar{p} - \bar{\pi}_{target}) \qquad (24)$$

### 2.7. Full Loss Formula

We define the autoregressive losses, to learn the full model and the early exit model as:

$$\mathcal{L}^{auto} = -\sum_{i=0}^{N} \log p_\theta(x_i | x_{<i}, a_{<i}) \qquad (25)$$

$$\mathcal{L}^{early} = -\sum_{i=0}^{N} \log p_\theta^{early}(x_i | x_{<i}). \qquad (26)$$

This gives our total loss calculation:

$$\mathcal{L} = \mathcal{L}^{auto} + \lambda_\pi \mathcal{L}^\pi + \lambda_{target} \mathcal{L}^{target} + \lambda_{early} \mathcal{L}^{early} \quad (27)$$

In our experiments, we set $\lambda_\pi = \lambda_{target} = 10^{-2}$ to encourage exploration and $\lambda_{early} = 10^{-1}$.

## 3. Related Work

### 3.1. Scaling Byte-Level Language Models with Tokenization Heuristics

Byte-level language models have recently been shown to scale competitively with tokenized transformers, with ByT5 (Xue et al., 2022) demonstrating that a pure byte-level architecture can match token-level performance, though at the cost of quadratic attention that makes long-sequence scaling impractical. Subsequent work has addressed this limitation along two main directions: leveraging subquadratic attention mechanisms (Wang et al., 2024; Zheng et al., 2025), and reducing sequence length through hierarchical or pooled representations. The latter includes fixed-stride downsampling approaches (Tay et al., 2022; Nawrot et al., 2022; Yu et al., 2023), as well as methods that segment byte streams at more semantically meaningful boundaries. Nawrot et al. (2023) first explored pooling at whitespace characters, later scaled by Slagle (2024) to outperform fixed-stride schemes. In parallel, Nawrot et al. also introduced entropy-based boundary detection (2023), leveraging spikes in next-byte entropy that correlate with semantic breaks; this idea was further scaled by Pagnoni et al. (2024), who showed that

entropy-guided downsampling can enable byte-level models to surpass token-level baselines at the $10^{22}$-FLOP scale.

While these methods demonstrate the promise of the autoregressive U-net architecture at scale and highlight some desirable properties that a tokenization mechanism for this architecture would have, all rely on heuristics, breaking the **End-to-end tokenizer training** desideratum in our setup.

### 3.2. Straight-Through Estimators for Learning Token Boundaries

Prior work on end-to-end tokenization has largely relied on straight-through estimators (STEs) to enable gradient-based optimization of token boundaries, in contrast to our use of a score-function estimator. These approaches relax the discrete boundary variables $a$ into continuous surrogates, differentiate through them, and then apply heuristic update rules to adjust $\pi_\theta(a)$ given the gradient $\frac{dL}{da}$. While such heuristics lack the theoretical guarantees enjoyed by score-function estimators §2.3, they have previously been proved effective in practice for learning mixture-of-experts (Shazeer et al., 2017) routing strategies, whereby there are too many decisions per token to provide a sufficiently low variance score function estimate.

The earliest demonstrations of STE-based boundary learning, such as Godey et al. (2022), showed that token boundaries can be learned for bidirectional encoders by introducing a non-causal pooling mechanism that injects a small amount of every byte's representation into each token representation. Although elegant, this technique is fundamentally incompatible with the causal constraints of contemporary autoregressive LLMs, and adapting STEs to handle counterfactuals of the form "what if this boundary were placed one byte later?" has proven difficult.

Subsequent work has extended STE strategies to autoregressive settings. Nawrot et al. (2023), for instance, employ a straight-through estimator built on segmentation heuristics introduced by Bhati et al. (2021). We compare to this method in our experiments §4. Kallini et al. (2025) propose a more robust scheme that performs a full forward pass over all bytes using only soft downsampling during training, discarding low-scoring bytes only at inference time; however, this violates our **Efficiency** desideratum, since it preserves the full computational cost of byte-level processing during training. Concurrent with our work, Hwang et al. (2025) pursue a related STE-based approach using specialized up- and downsampling modules, which themselves introduce heuristic design choices—for example, initializing the downsampler to favor boundary placement at dissimilar byte transitions (Main Horse [pseudonym], 2025). Together, these methods highlight both the promise and the limitations of STEs for token-boundary learning, motivating alternative estimators with stronger theoretical footing.

We would like to highlight concurrent work by Wu et al. (2026) which independely proposed a score-function estimator to learn token boundaries for time series data under the framework of reinforcement learning. We also highlight further innovations on the autoregressive U-net architecture in Appendix C.

## 4. Experiments

To evaluate our proposed score-function–based approach, we train autoregressive transformers on a filtered subset of the `FineWeb` dataset (Penedo et al., 2024), where we keep sequences of at least 4096 bytes. Subsequently, we truncate these sequences to exactly 4096 bytes. All models have approximately 147 million parameters and are trained with a target downsample rate of $\frac{1}{5}$. We select dataset size, batch size, and learning rate using the scaling laws derived in Porian et al. (2024). See Appendix D for further experimental details and hyperparameters.

We compare our score-function estimator against four further tokenization strategies: a uniform baseline that places exactly $4096 \, \bar{\pi}_{target} = 819$ evenly spaced boundaries, the straight-through estimators of Nawrot et al. (2023) and Hwang et al. (2025), and a *BPE-guidance* baseline (§4.3) that fixes the token boundaries to those induced by a trained BPE tokenizer rather than learning them.

Because our models are several orders of magnitude smaller in training compute than contemporary frontier LLMs, the global downstream effects of learned tokenization on language modeling quality are difficult to assess directly. For example, the recently reported superiority autoregressive U-Nets over purely token-based architectures (Pagnoni et al., 2024; Hwang et al., 2025) was only observed to emerge at the $> 10^{21}$-`Flop` scale. For these reasons, our analysis focuses on qualitative properties of the learned token boundaries rather than downstream perplexity.

We publicly release our code at `https://github.com/SamD770/bitter-lesson-tokenization`

### 4.1. Learned Tokenization Strategies for Natural Language

We train 147-million parameter models on our filtered version of `FineWeb` .

Figure 2 illustrates the token boundaries produced by our method on held-out `FineWeb` samples. Remarkably, despite having no inductive bias toward linguistic structure, the model reliably learns to place boundaries at whitespace-like characters, such as "\n" and " ". Additional samples are provided in Appendix B.1. A comparison to the tokenization patterns learned by Nawrot et al. (2023) is provided in B.2 .

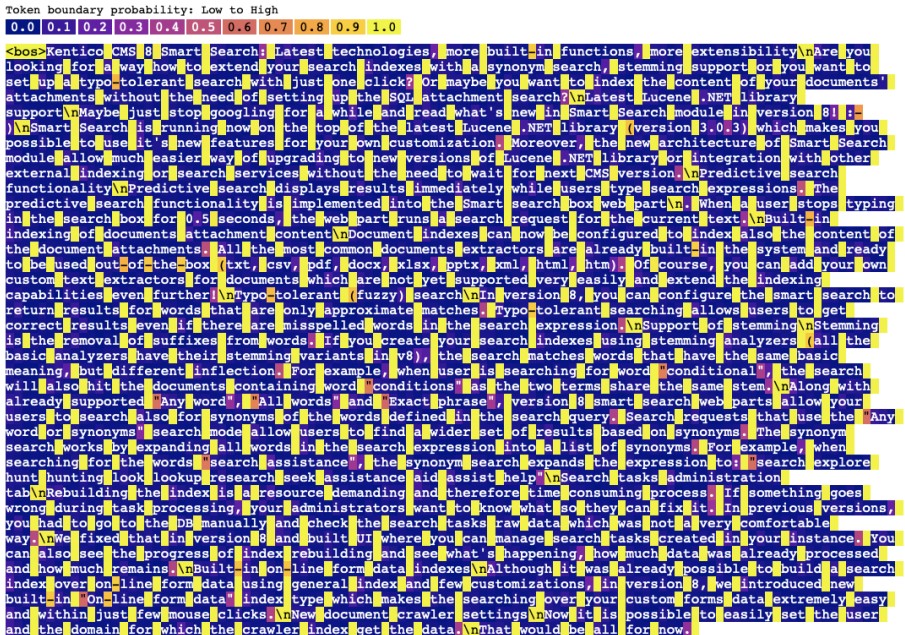

*Figure 2.* Token boundaries learned by our 147M-parameter model on a held-out sample of the `FineWeb` dataset. Yellow and blue characters characters indicate high or low values of $\pi_\theta(a)$ respectively at the corresponding bytes.

In appendix B.6, we further study the sensitivity of the learned tokenization strategy to the target downsample rate by varying the *tokenization aspect ratio:* we train models with $n = 2, 4, 6$ token-level transformer layers (sizes ranging from 20 to 40-million parameters) and a target downsample rate of $\bar{\pi}_{target} = \frac{1}{n}$, keeping the FLOPs -per-sequence fixed. The $n = 2$ model converges to a tokenization strategy of allocating almost exactly two bytes per token, with the exception of also drawing token boundaries at periods. By contrast, the $n = 4, 6$ models again discover whitespace-aligned boundaries and exhibit with varying degrees of chunking of longer words.

### 4.2. Natural Language Performance

In Figure 3, we plot validation loss curves on our filtered `FineWeb` for our model compared to the straight-through estimator proposed by Hwang et al, Nawrot et al. and a random baseline. Following Kaplan et al. (2020), we estimate the number of FLOPs in the model training run as approximately the number of FLOPs used for the matrix parameters in the forward/backward pass (which is the dominant source for LLMs), which takes a value of 6 FLOPs per parameter per byte or token, the latter depending on which level the layer operates on (Bahdanau, 2022). We assume that the embedding operation is performed using an efficient lookup. Small deviations in FLOPs per batch for each method occur as a result of variations in the max token sequence length in the batch.

We observe the qualitatively more semantically meaningful token boundaries that our method finds over the baseline straight-through estimates to translate to a consistent improvement in validation loss over the training run. In Table 1 we compare methods across a range of downstream natural language understanding benchmarks (Bisk et al., 2020; Zellers et al., 2019; Clark et al., 2018; Paperno et al., 2016). At the 147M scale, zero-shot accuracy on these benchmarks is close to chance and dominated by noise, making comparison across methods unreliable; we therefore report the bits-per-byte that each model assigns to the correct continuation, which is a substantially lower-variance signal. Among the four primary methods, our learned policy attains the lowest bits-per-byte on the held-out `FineWeb` test set and on `PIQA`, `HellaSwag` and `LAMBADA`, with the remaining differences (notably on `ARC-Easy`) lying within roughly one standard error. We additionally report a variant in which the boundary-logit window is reduced to $w = 1$ (§2.5); this ablation performs comparably to our full model, indicating that the additional sliding-window context is not necessary for strong performance at this scale.

### 4.3. Comparison to BPE-Guided Downsampling

For natural language, we compare the learned token boundaries of each method to a *BPE-guidance* baseline that fixes the token boundary decisions based on a BPE tokenizer trained on `FineWeb` (with an effective 200k-token vocabulary, giving a downsampling rate of 0.207). We leave the autoregressive U-Net architecture otherwise unchanged. As

*Table 1.* Performance of 147M parameter models on downstream tasks. We report the bits-per-byte (a length-normalized cross-entropy; *lower is better*) that each model assigns to the correct continuation, rather than zero-shot accuracy: at this scale accuracy is close to chance and high-variance, whereas this measure is substantially lower-variance (see Appendix D). `FineWeb Test` is the bits-per-byte on held-out text. Best result per column in **bold**; $\pm$ denotes the standard error.

|  | PIQA | HellaSwag | ARC-Easy | LAMBADA | FineWeb Test |
|---|---|---|---|---|---|
| Uniform | 1.660 ±0.011 | 1.306 ±0.002 | **1.974** ±0.017 | 1.926 ±0.012 | 1.376 ±0.003 |
| Dynamic (Nawrot et al.) | 1.737 ±0.010 | 1.340 ±0.002 | 2.011 ±0.017 | 1.956 ±0.012 | 1.372 ±0.003 |
| H-Net (Hwang et al.) | 1.641 ±0.011 | 1.313 ±0.002 | 2.000 ±0.017 | 2.130 ±0.012 | 1.386 ±0.003 |
| BPE guidance | 1.589 ±0.011 | 1.230 ±0.002 | 2.084 ±0.019 | 1.645 ±0.013 | 1.299 ±0.003 |
| Ours ($w = 1$) | 1.561 ±0.011 | **1.199** ±0.002 | 1.987 ±0.018 | **1.584** ±0.013 | **1.280** ±0.003 |
| Ours | **1.557** ±0.011 | 1.212 ±0.002 | 2.016 ±0.018 | 1.737 ±0.013 | 1.297 ±0.003 |

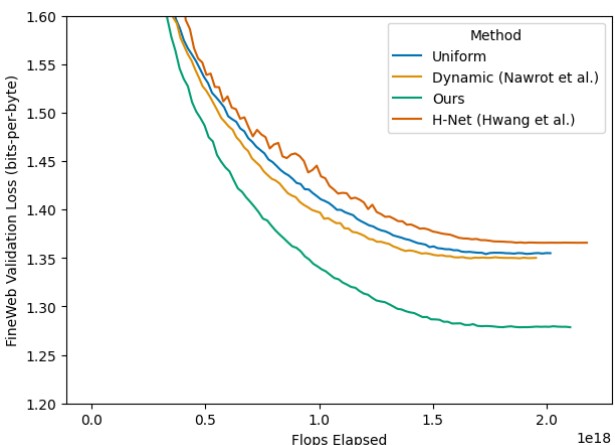

*Figure 3.* Flops vs validation loss curves for 147M parameter models, trained on `FineWeb`, measured in bits-per-byte. Uniform random in blue, Straight through estimates in orange, ours in green. Validation loss values at $1.95 \times 10^{18}$ `FLOPs` are 1.355, 1.350, 1.36 and 1.279 bits-per-byte respectively

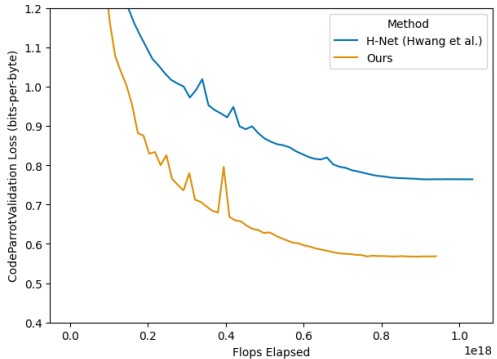

*Figure 4.* Flops vs validation loss curves for 90M parameter models, trained on `CodeParrot`, measured in bits-per-byte. H-Net in blue ours in orange. Validation loss values at $0.95 \times 10^{18}$ `FLOPs` are 0.769 and 0.568 bits-per-byte respectively

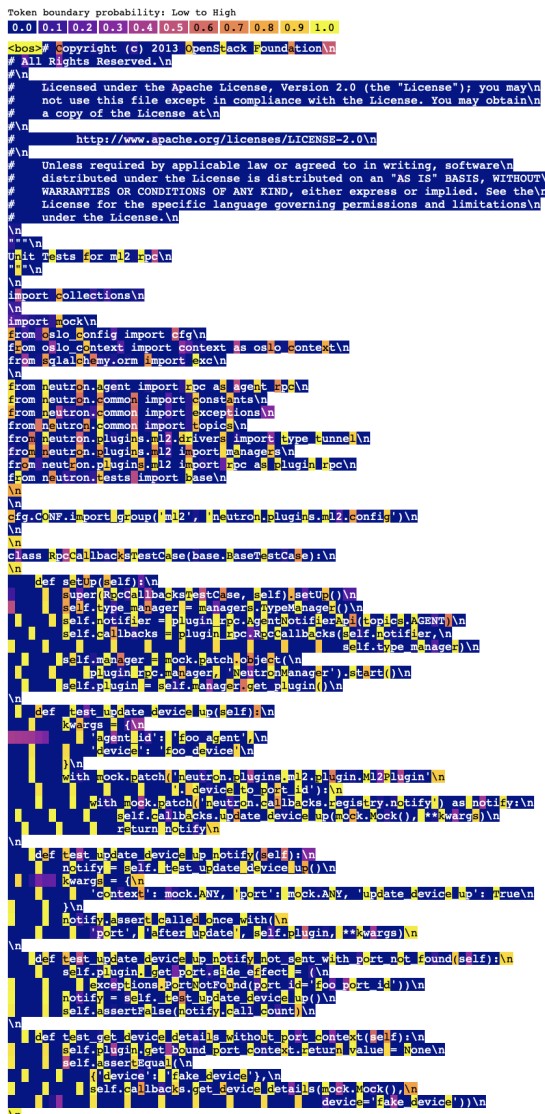

*Figure 5.* Token boundaries learned by our 90M-parameter model on a held-out sample of the `CodeParrot` dataset. Yellow and blue characters characters indicate high or low values of $\pi_\theta(a)$ respectively at the corresponding bytes.

the resulting boundaries leak information about future bytes, we apply a single right shift to the boundary indicators before use; we detail this construction in Appendix D.1. As reported in Table 1, BPE guidance attains a `FineWeb` test bits-per-byte of 1.299, essentially identical to the 1.297 of our learned policy. Our method is thus the only dynamic tokenization strategy we evaluate that recovers the performance of BPE-guided downsampling without access to these external priors.

### 4.4. Python Code

We train 90 million parameter models on the `CodeParrot` dataset (Hugging Face, 2022) of python code, using both our method and the straight-through estimate of Hwang et al (2025). In Figure 5 we report that our model learns to draw token boundaries at the beginning of module names, space tokens to contain at least 2 bytes, and learns to note expend test-time compute on the Apache License, which is frequently repeated throughout the training dataset. Additional samples are provided in Appendix B.4 and tokenization strategies learned by H-Net are provided in appendix B.5. In Figure 4 we plot the validation loss for both methods, finding that this intuitively appealing tokenization strategy leads to an improved downstream loss.

## 5. Conclusion

We propose a flexible parameterization of the problem of intra-architecture tokenization, which generalises prior proposed heuristics for this problem. We have demonstrated that applying our variance reductions yields a score function estimator that can optimize this parameterization to learn semantic boundaries §4.1 in text soley from optimizing the cross entropy and that doing so outperforms prior straight-through estimation based techniques §4.2. We hope that score function estimates become the method of choice for tokenization as models become increasingly end-to-end.

As we continue this work, we look to compare to concurrent work (Hwang et al., 2025) and extend our experiments to more diverse text data. More investigation is also needed into finding the optimal tokenization aspect ratios. We pose the following open problems: (1) At the 20-40 million parameter scale, we found that the aspect ratio of 2 produced the best loss, how does the optimal tokenization aspect ratio scale with model size? (2) Our initial attempts to analyse downsample rate scaling in a compute-efficient manner via training a single model to accept a variable target downsample rate (Beyer et al., 2023) resulted in a model which learned a more robust but less effective tokenization strategy of drawing many token boundaries at important parts of the sequence; can our results learning effective strategies on fixed downsample rates be generalised in this way?

## Impact Statement

This paper presents work whose goal is to advance the field of Machine Learning. There are many potential societal consequences of our work, none which we feel must be specifically highlighted here. That said, by reducing reliance on fixed, language-specific tokenization schemes, end-to-end and byte-level tokenization approaches may help improve the inclusivity and performance of language models for non-English and underrepresented languages.

## Acknowledgements

We thank Abigail See, Frédéric Berdoz, and Maximilian-David Rumpf for their careful reading of early drafts of this paper and for their valuable feedback, which substantially improved the clarity and presentation of this work.

# A. Proofs

## A.1. Loss Breakdown

$$
\begin{aligned}
\nabla_\theta \mathbb{E}_{a\sim\pi_\theta} \log p_\theta(y|a,x) =& \nabla_\theta \sum_a \log p_\theta(y|a,x)\pi_\theta(a|x) \\
=& \sum_a \left(\nabla_\theta \log p_\theta(y|a,x)\right) \pi_\theta(a|x) \\
& + \sum_a \log p_\theta(y|a,x) \left(\nabla_\theta \pi_\theta(a|x)\right) \\
=& \nabla_\theta \sum_a \left(\nabla_\theta \log p_\theta(y|a,x) + \log p_\theta(y|a,x)\nabla_\theta \log \pi_\theta(a|x)\right) \pi_\theta(a|x) \\
=& \mathbb{E}_{a\sim\pi_\theta} (\underbrace{\nabla_\theta \log p_\theta(y|a,x)}_{\text{cross entropy loss}} + \underbrace{\log p_\theta(y|a,x)\nabla_\theta \log \pi_\theta(a|x)}_{\text{REINFORCE gradient}})
\end{aligned}
$$

## B. Tokenization Strategies

In the following plots, multi-byte characters, such as ü are rendered using the probability of the last byte representing the character.

### B.1. Our Method

### B.2. Straight-Through Estimator from Nawrot et al. (Nawrot et al., 2023)

### B.3. Hnet (Hwang et al.) (Hwang et al., 2025)

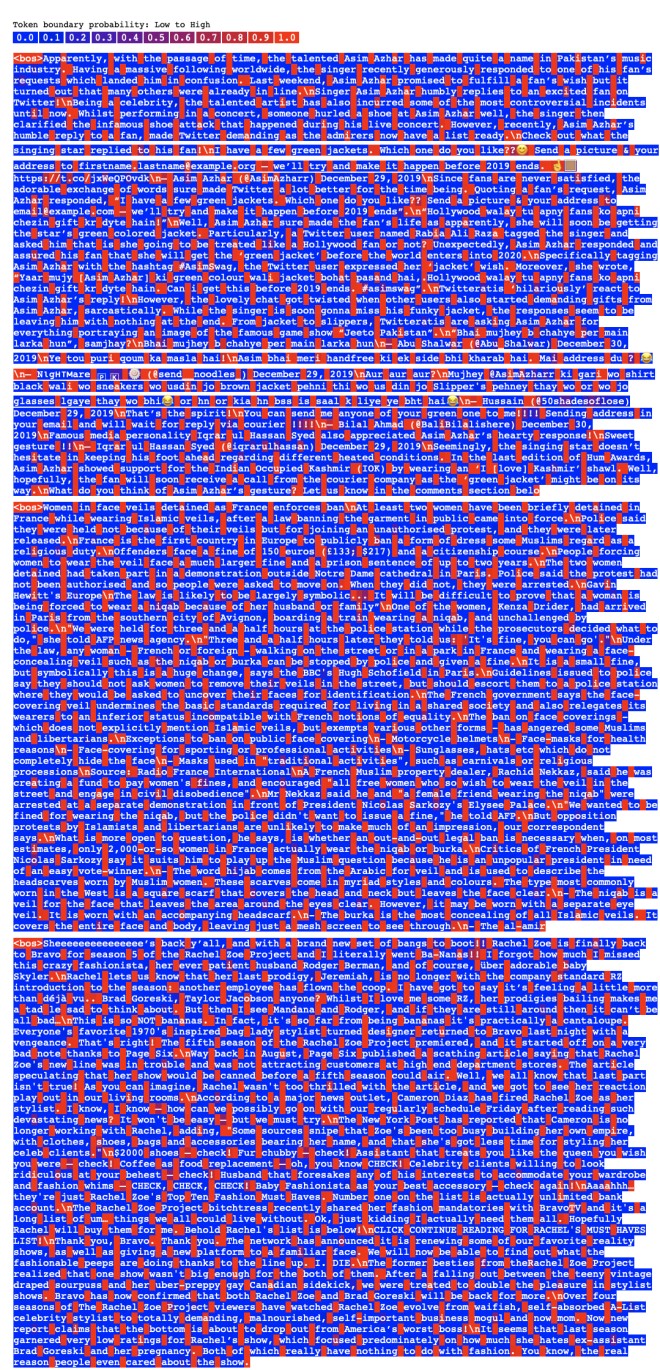

*Figure 6.* Token boundaries learned by our 147M-parameter model on a held-out sample of the FineWeb dataset. Red and blue characters characters indicate high or low values of $\pi_\theta(a)$ respectively at the corresponding bytes.

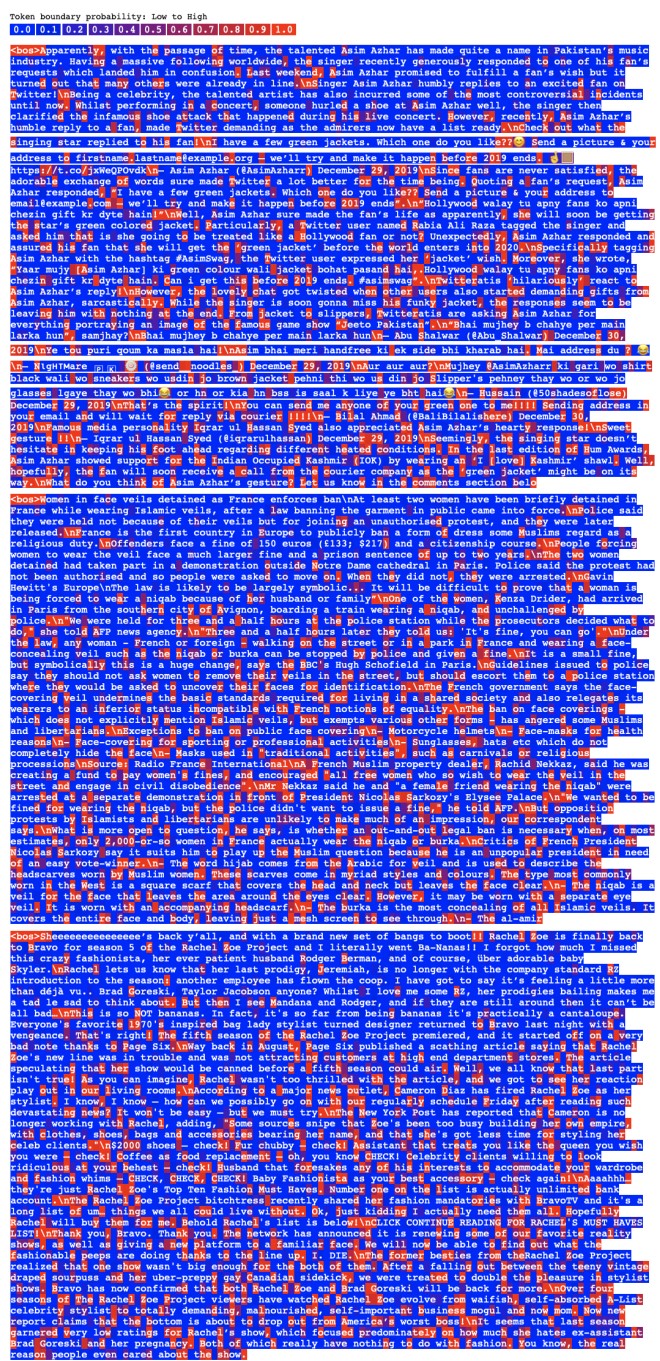

*Figure 7.* Token boundaries learned by a 147M-parameter model using the straight-through estimator of Nawrot et al. (Nawrot et al., 2023), on held-out samples of the FineWeb dataset. Instead of the probabilities, here we plot the soft boundaries at the output of the Gumbel-Sigmoid. Red and blue characters characters indicate high or low values of $\hat{b}_t$ respectively at the corresponding bytes.

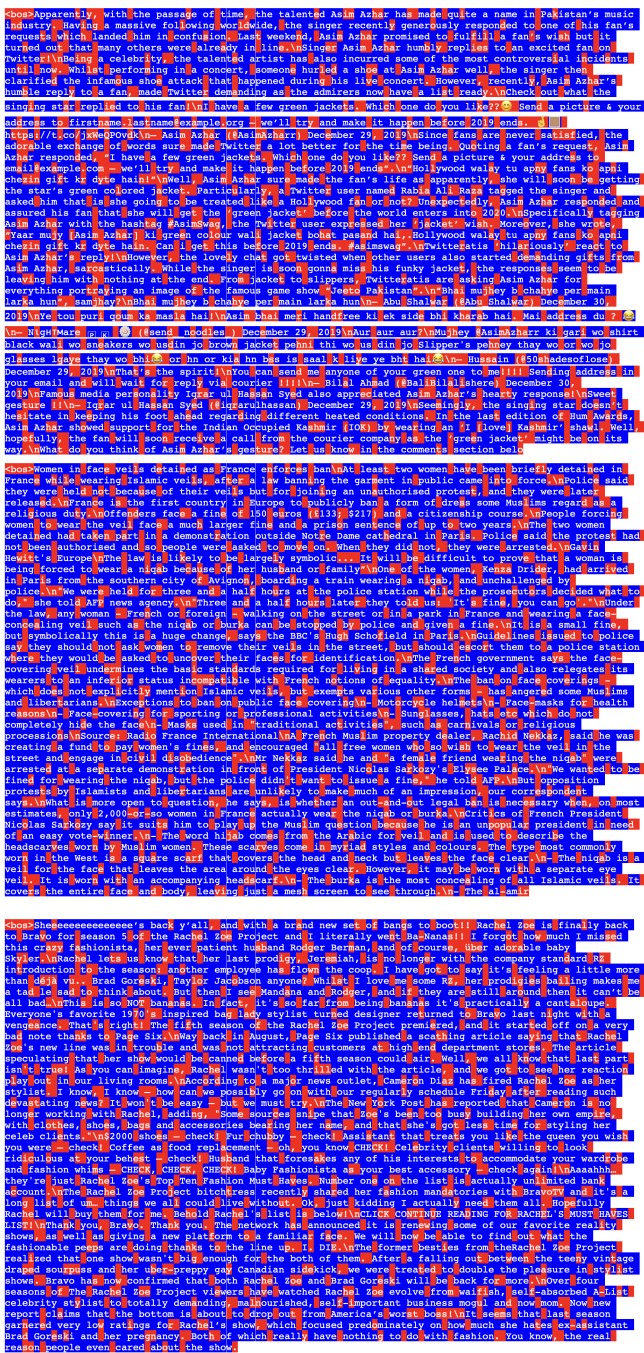

Figure 8. Token boundaries learned by a 147M-parameter model using the straight-through estimator of Hwang et al. (Hwang et al., 2025), on held-out samples of the FineWeb dataset. Instead of the probabilities, here we plot the hard token boundaries. Red and blue characters characters indicate high or low values of $\hat{b}_t$ respectively at the corresponding bytes.

*Figure 9.* Token boundaries learned by our 90M-parameter model on a held-out sample of the `CodeParrot` dataset. Red and blue characters characters indicate high or low values of $\pi_\theta(a)$ respectively at the corresponding bytes.

## B.4. Our Method on `CodeParrot`

*Figure 10.* Token boundaries learned by a 90M-parameter model using the straight-through estimator of Hwang et al. (Hwang et al., 2025), on held-out samples of the CodeParrot dataset. Instead of the probabilities, here we plot the hard token boundaries. Red and blue characters characters indicate high or low values of $\hat{b}_t$ respectively at the corresponding bytes.

## B.5. H-Net (Hwang et al.) (Hwang et al., 2025) on CodeParrot

## B.6. Varying the Downsample Rate

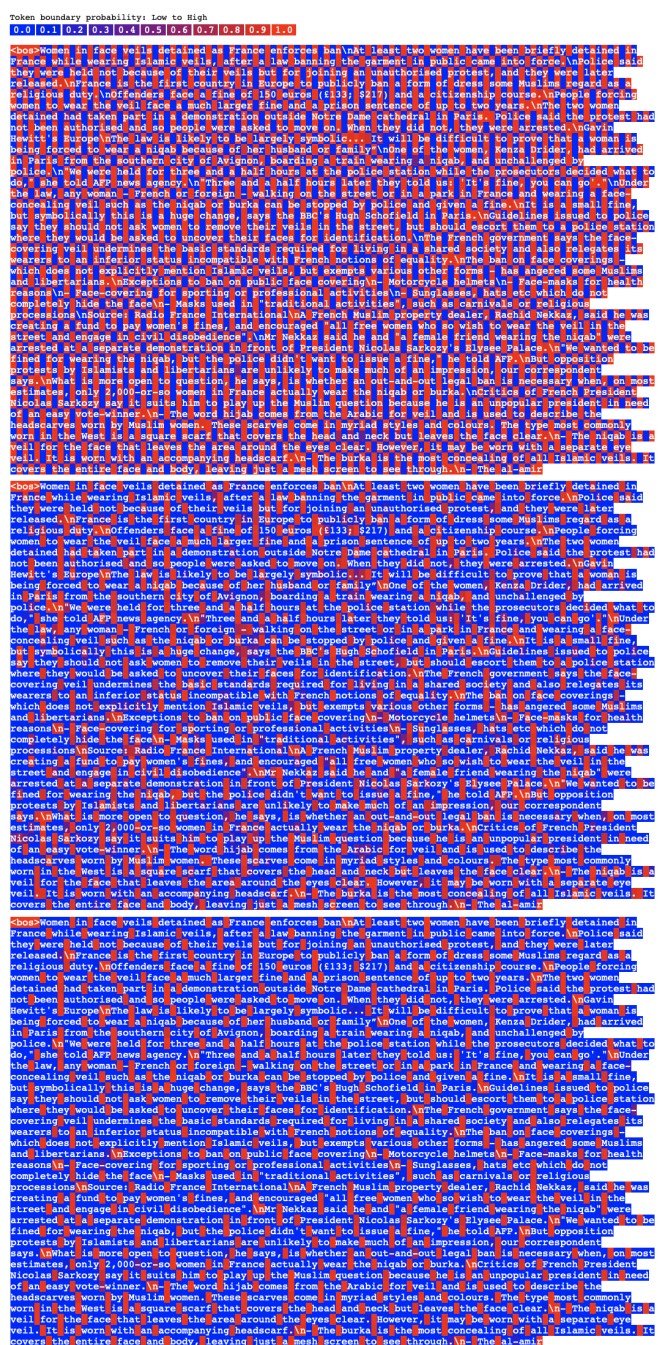

*Figure 11.* Token boundaries learned by an array of models using our method with varying target downsample rates $\bar{\pi}_{target} = \frac{1}{2}, \frac{1}{4}, \frac{1}{6}$ (top, middle, bottom, respectively) on a held-out sample of the `FineWeb` dataset. Red and blue characters characters indicate high or low values of $\pi_\theta(a)$ respectively at the corresponding bytes.

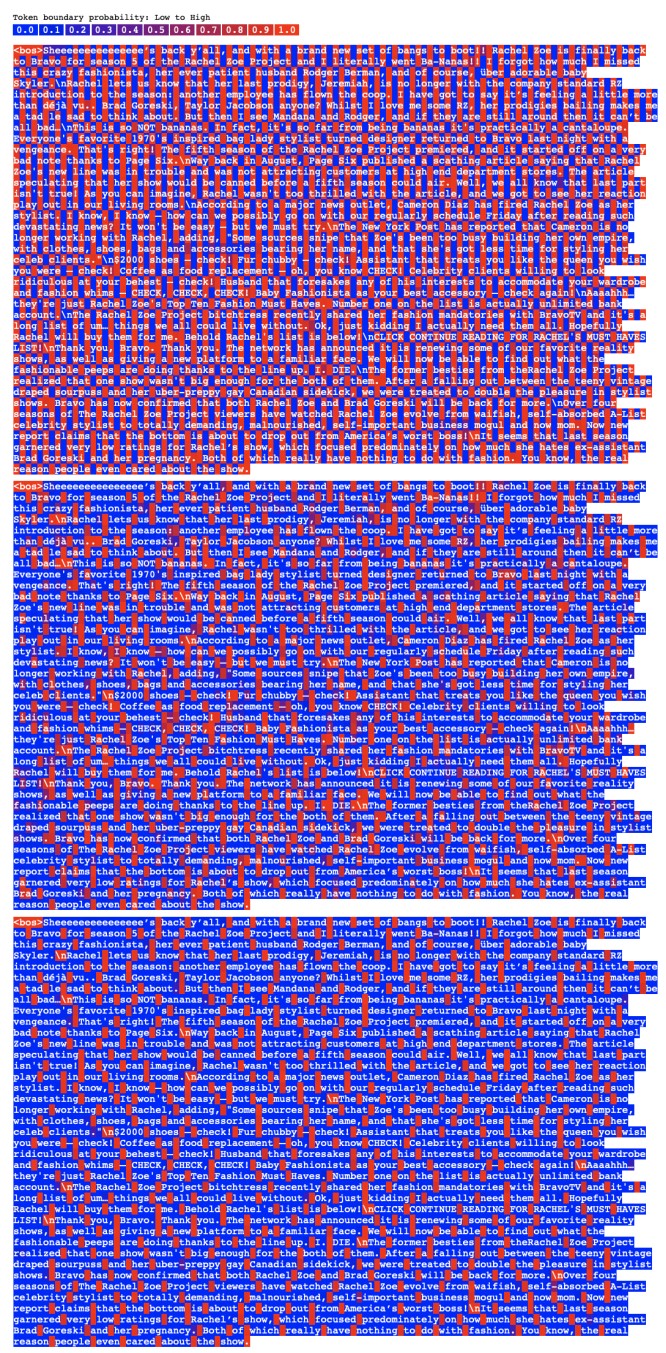

Figure 12. Token boundaries learned by an array of models using our method with varying target downsample rates $\bar{\pi}_{target} = \frac{1}{2}, \frac{1}{4}, \frac{1}{6}$ (top, middle, bottom,) of on a held-out sample of the FineWeb dataset. Red and blue characters characters indicate high or low values of $\pi_\theta(a)$ respectively at the corresponding bytes.

# C. Architectural Innovations in Prior Work

We focus on the method of deciding token boundaries, and thus use a simple architecture for the purpose of not confounding our experiments. Nonetheless, we would like to highlight the following architectural innovations that have been proposed to improve autoregressive U-nets:

- Using a vocabulary of byte-level $n$-grams which are added to the input embeddings to improve information flow (Pagnoni et al., 2024).

- Using cross-attention between the byte-level and token-level transformers (Pagnoni et al., 2024).

- Using subquadratic sequence-to-sequence blocks at the byte level (Hwang et al., 2025).

- Using multiple levels of tokenization (Hwang et al., 2025).

- Smoothing between tokenization levels in upsampling (Hwang et al., 2025).

We would like to note that all of these methods could be used with our method to learn token boundaries.

| Model size | 147M | 90M |
|---|---|---|
| `embedding_dim` | 768 | 768 |
| `num_heads` | 12 | 12 |
| `n_down_layers` | 4 | 4 |
| `n_mid_layers` | 12 | 6 |
| `n_up_layers` | 4 | 4 |
| `learning_rate` | $1.5 \times 10^{-3}$ | $2 \times 10^{-3}$ |
| `effective_batch_size` | 128 | 128 |
| `warmup_bytes` | $4.5 \times 10^8$ | $2.5 \times 10^8$ |
| `training_bytes` | $6 \times 10^9$ | $3.4 \times 10^9$ |
| `downsample_rate_target` | $\frac{1}{5}$ | $\frac{1}{5}$ |

*Table 2.* The hyperparameters used for our runs varying the token boundary selection method used on natural language (left) python code (middle) and the varying tokenization aspect ratio (right)

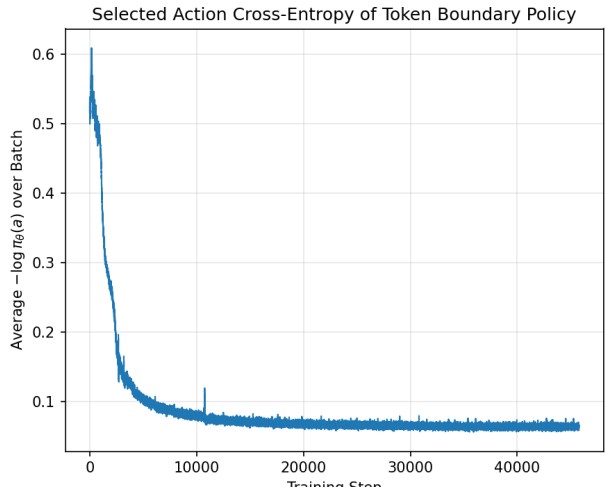

*Figure 13.* The mean of $-\log p_\theta(a_t|a_{<t})$ over batches across the training run. We observe that boundaries stabilize after approximately 10% of the training run.

| Method | Downsampling rate |
|---|---|
| H-Net (Hwang et al.) | 0.206 |
| Dynamic (Nawrot et al.) | 0.200 |
| Uniform | 0.200 |
| Ours | 0.204 |
| BPE guidance | 0.207 |

*Table 3.* Downsampling rates at convergence ($\bar{a}$) on `FineWeb` for the different methods.

## D. Further Experimental Details

We train models on a total of `training_bytes` bytes with a cosine learning rate schedule with a warmup of `warmup_bytes` and a maximum learning rate of `learning_rate`. We use the AdamW optimizer with default parameters $(\beta_1, \beta_2) = (0.9, 0.999)$. Per gradient update, we performa a forward & backward pass on `effective_batch_size` sequences of length 4096 .

We use a decoder-only transformer architecture, with `n_down_layers` and `n_up_layers` byte-level transformer decoder layers with sliding window attention with window size 64 before and after the token-level backbone which consists of `n_mid_layers` token-level transformer decoder layers with full causal attention. All attention layers have `num_heads` heads. We make the model dimension the same for byte and token-level layers, such that $d_{enc} = d_{mid} = d_{dec} =$ `embedding_dim`. Byte-level and token-level layers have an MLP hidden dimension of `embedding_dim` and $4 \times$ `embedding_dim` respectively, making the flops-per-layer roughly consistent. We closely follow the transformer architectural choices of Gemma 2 (Rivière et al., 2024), with GeGLU non-linearity, Rotary Position Embeddings, post and pre- RMSNorm and Logit soft-capping.

See table 2 for the hyperparamters used in our 147-M parameter experiment and our experiments varying the token aspect ratio.

For each multiple-choice benchmark we score every candidate answer by the total next-byte cross-entropy the model assigns to its bytes, normalized by the answer length in bytes (bits-per-byte); we report this value for the gold answer. For LAMBADA we score the final word, and for `FineWeb Test` the full held-out sequence. We prefer this to zero-shot accuracy because, at the 147M scale, accuracy is close to chance and its variance across runs swamps the differences

between methods.

While training `FLOPs` are held constant across all methods, inference `FLOPs`—which depend on the downsampling rate—also affect benchmark scores. Table 3 reports the downsampling rates at convergence on `FineWeb` for each method. Our method achieves a slightly lower downsampling rate than both BPE guidance and H-Net, meaning that we give a mild advantage to those baselines rather than benefiting from an inconsistent comparison.

### D.1. BPE-Guidance Baseline

For the BPE-guidance baseline of §4.3 we train a BPE tokenizer with a vocabulary of 200k tokens on the `FineWeb` training set, yielding a downsampling rate of 0.207, and draw a token boundary at the final byte of each BPE token. A subtlety is that BPE boundaries leak information about future bytes: because, for example, `"tokens"` is encoded as a single token whereas `"tokenization"` is split as `"token"|"ization"`, a boundary placed after the `"n"` of `"token"` already reveals that the next byte is not `"s"`. Since our architecture exposes $a_i$ when predicting byte $x_{i+1}$,

using the raw boundaries would grant the model an unfair lookahead. We remove this leakage by applying a single right shift to the boundary indicators, so that each boundary is revealed one byte after the token it terminates. For instance, `"hello␣tokenization"` receives the shifted indicators:

```
bytes:   h e l l o ␣ t o k e n i z a t i o n
shifted: 0 0 0 0 0 1 0 0 0 0 0 1 0 0 0 0 0 0
```

All other hyperparameters match the 147M runs of §4.2.

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
