# OpenReview forum: "You Can Learn Tokenization End-to-End with Reinforcement Learning"
_ICML.cc/2026/Conference — ICML 2026 regular_

### Official Review · Reviewer_TUGD · 2026-03-10

**Soundness:** 2
**Presentation:** 2
**Significance:** 3
**Originality:** 3
**Overall Recommendation:** 4
**Confidence:** 5

**Summary:**

The paper focuses on learning tokenization boundaries for language models. While many works, including old and recent ones, mainly use STE, this paper explores using reinforcement learning by modeling the problem as a sequential decision process and optimizing it together with the cross-entropy loss in an end-to-end manner. To make learning efficient and effective, it further proposes using the early exit relative reward and time discounting. Moreover, a downsampling rate targeting loss is used, which affects the efficiency and accuracy by encouraging the tokenization to move towards a certain downsampling rate. Experiments are mainly conducted on a 100M model scale with comparisons to some recent methods. The paper also shows the tokenization boundaries detected by the proposed method on web text and code samples.

**Compliance With Llm Reviewing Policy:**

Affirmed.

**Final Justification:**

I'd like to keep my current score (weak accept), but I don't think my concerns are fully addressed. I think the general idea of incorporating RL into tokenization is very interesting and the equations look correct (though the design has been somewhat complicated in the last stage). However, the paper still finally does not show better downstream task performance by training the small model (147M) longer and with the correct experimental setup. I don't fully buy the excuse of the computational resource, given the whole rebuttal period is almost two weeks. Also training such a small model with a longer training e.g., 100B tokens is not a very hard problem, especially compared to other architecture design papers.

**Key Questions For Authors:**

Please see above.

**Limitations:**

yes

**Strengths And Weaknesses:**

Soundness
1. I follow the theoretical equations one by one and they look good and solid to me. One small typo: it seems that line 617 wrongly adds a gradient operator.
2. Experiments do not sound very solid to me. For example, the authors mention that they use truncation to match the sequence length. However, a more widely used technique is to concatenate examples together so that the web text can be fully utilized. There also seems to be a lack of ablation studies for determining the coefficient in Eq. (27). Are the coefficients (different lambdas) sensitive to the accuracy performance? Also, in Table 1, given that PIQA is a binary-choice task and ARC-Easy and HellaSwag are four-choice tasks, the overall performance looks close to random guessing, which makes it difficult for readers to understand the performance differences across models. While I understand that this might be due to restricted compute resources, it would be better to scale the model size (e.g., 300M) or the number of training tokens (e.g., 20B) slightly to provide better validation.
3. While the paper claims that the proposed method has tighter theoretical guarantees than those STE ones, I did not find rigourous proof across the paper.

Presentation
1. The presentations of Figures 3 and 4 are not clear.
2. While the equations are clear, to make readers understand them more easily, it would be helpful if authors could give more intuition before deriving these formulas and also provide some summarizations before going into details.

Significance
1. The paper is tackling an important problem and pioneeringly introduces reinforcement learning into deciding tokenization boundaries, compared to those of the STE methods, which is significant.

Originality
1. The method is novel, as it formulates the cross-entropy loss and strategy loss together in an end-to-end manner. Moreover, to make it practical, the paper proposes using an early-exit reward and time discounting for easier optimization.
2. The method includes a loss called downsample rate targeting. Although it introduces another loss term in the optimization, I see this as a mechanism to control the final efficiency, which other methods do not provide. For example, larger targets indicate more compression and thus higher efficiency. It would be helpful if the authors could provide more discussion on different target rates in terms of both accuracy and efficiency.

---

> ### Author Rebuttal · Authors · 2026-03-31
>
> Thank you for the careful review, particularly the thorough verification of our theoretical derivations. We address each point below.
>
> **Typo in line 617.** We thank the reviewer for catching this. The erroneous gradient operator has been removed in the revised manuscript.
>
> **Sequence packing.** We agree that sequence packing (i.e., concatenating examples) should be used in production settings. We opted for truncation in our experiments for simplicity of our training infrastructure.
>
> **Near-random performance on downstream benchmarks.** We acknowledge that the zero-shot accuracy at this model scale is close to random guessing, which makes it difficult to draw meaningful comparisons across models. Unfortunately, scaling further is not within our compute budget. To reduce noise and provide a more informative comparison, we have also computed the bits-per-byte for the correct answer in each dataset for all models. These results are included in the revised manuscript.
>
> | Model | ARC-Easy | HellaSwag | LAMBADA | PIQA | FineWeb Test |
> |---|---|---|---|---|---|
> | H-Net (Hwang et al.) | 2.000 ±0.017 | 1.313 ±0.002 | 2.130 ±0.012 | 1.641 ±0.011 |  1.386 ± 0.003 |
> | Dynamic (Nawrot et al.) | 2.011 ±0.017 | 1.340 ±0.002 | 1.956 ±0.012 | 1.737 ±0.010 |  1.372 ± 0.003 |
> | Uniform | **1.974 ±0.017** | 1.306 ±0.002 | 1.926 ±0.012 | 1.660 ±0.011 |  1.376 ± 0.003 |
> | Ours | 2.016 ±0.018 | **1.212 ±0.002** | 1.737 ±0.013 | **1.557 ±0.011** | **1.297 ± 0.003** |
> | BPE guidance | 2.084 ±0.019 | 1.230 ±0.002 | **1.645 ±0.013** | 1.589 ±0.011 | 1.299 ± 0.003 |
>
> **Clarity of Figures 3 and 4.** We have updated the axis labels and increased the size of these figures to improve legibility.
>
> **Intuition for equations.** We appreciate this suggestion and have added more intuitive explanations before the key derivations, as well as brief summaries of the main takeaways, to improve readability.
>
> **Theoretical guarantees relative to STE.** We have added formal derivations to Appendix A showing that ignoring prior rewards, using early-exit relative rewards, and using batch-relative advantages all yield consistent estimators of the score function.
>
> **Sensitivity to loss coefficients.** We refer the reviewer to our ablation study (described in detail in our response to Reviewer XMsP), where we systematically ablate the key components of our method. Notably, removing the consistency loss ($\\lambda_{\\text{target}} = 1$) causes collapse to a trivial solution, while the other components have more moderate effects at this scale.
>
> **Effect of different target downsampling rates.** For our small-scale experiments, we observed that smaller tokens (less aggressive downsampling) tend to yield better performance. However, scaling the downsampling rate beyond what is achievable with pure BPE still produced meaningful and non-trivial token boundaries. We believe a more systematic exploration of target rates at larger scales would be valuable future work.

---

> > ### Author Rebuttal · Reviewer_TUGD · 2026-04-02
> >
> > Thanks for the reply. I would be happy to keep my positive attitude. However, given that two of my core questions about downstream acc with overtraining models and efficiency discussion of different downsampling rates, I won't consider raising up the score for now.

---

> > > ### Author Response · Authors · 2026-04-08
> > >
> > > To respond to your query with respect to the hyperparameters: "...Are the coefficients (different lambdas) sensitive to the accuracy performance?" We train 40M-parameter models, allowing $\lambda_{\\pi}$ and  $\lambda_{target}$ to vary 4 orders of magnitude (between 1. and 0.001) and $\lambda_{early}$ to vary 3 orders of magnitude (1. and 0.01). For each run, the other hyperparameters are given the same values as in Section 2. Giving comparable evaluations is tricky due to a strongly differing downsampling rates and a weaker signal at this model size, but we give qualitative results here: https://anonymous.4open.science/r/40M_scaled_ablations_renders-C85C . Overall, we note the following trends:
> > >
> > > - Allowing $\\frac{\\lambda_{target}}{\\lambda_{\\pi}} = 10.$ leads to a comparable to baseline but less sharp distribution for token boundaries, and letting $\\frac{\\lambda_{target}}{\\lambda_{\\pi}} = 100.$ leads to the model staying with uniformly random token boundaries.
> > >
> > > - Conversely, $\\frac{\\lambda_{target}}{\\lambda_{\\pi}} = 0.1$ leads to a comparable result as the baseline, but allowing  $\\frac{\\lambda_{target}}{\\lambda_{\\pi}} = 0.01$ leads to a much higher downsampling rate ($\\approx 0.35$).
> > >
> > > - $\lambda_{early} = 1.$ leads to a collapse in quality, but $\lambda_{early} = 0.01$ produces similar results as to the baseline.
> > >
> > > To conclude, the $\\lambda_{target}$ and $\\lambda_{\\pi} = 0.1$ can be modified within 1 OOM of the reported values and still give good qualitative results, $\lambda_{early}$ can be reduced or set to zero altogether (see other ablations provided to reviewer XMsP) and not affect the baseline.

---

### Official Review · Reviewer_fpf9 · 2026-03-11

**Soundness:** 2
**Presentation:** 3
**Significance:** 3
**Originality:** 3
**Overall Recommendation:** 3
**Confidence:** 3

**Summary:**

The paper proposes an end-to-end tokenization method for Large Language Models (LLMs) by formulating the discrete token boundary selection as a reinforcement learning problem . Bypassing heuristic boundary rules and straight-through estimators (STEs) , the authors employ a score-function estimator (REINFORCE) to optimize token routing inside an autoregressive U-Net architecture . To mitigate the notoriously high variance of policy gradients, they introduce time discounting, batch-relative advantage centering, and an early-exit baseline reward . Evaluated at the 147M and 90M parameter scales on FineWeb and CodeParrot subsets , the model autonomously discovers semantic boundaries (e.g., whitespaces) and achieves slightly better validation losses than comparable STE baselines

**Compliance With Llm Reviewing Policy:**

Affirmed.

**Key Questions For Authors:**

1. The paper claims to eliminate human-crafted heuristics for pure end-to-end learning. However, to prevent the model from assigning a token boundary to every byte, you impose an artificial target downsample rate, $\overline{\pi}_{target}$ (e.g., 1/5), and apply a penalty loss . Does this not imply that the model is merely performing constrained optimization under human-imposed shackles rather than freely discovering language rules? How is this global constraint fundamentally different from the heuristic rules you aim to eliminate?
2. In the core results (Table 1), the baselines only include Uniform chunking and peer methods (Nawrot et al., Hwang et al.). You deliberately avoided comparisons with highly optimized traditional BPE tokenizers (like the OpenAI tiktoken standard). Without this absolute gold standard, how can readers assess whether this highly complex, compute-intensive RL method yields any substantive performance or business improvements over simple, mature BPE?
3. As sequence lengths and model dimensions increase, the variance of score-function estimators typically explodes. Can you provide empirical measurements of the gradient variance for $\pi_\theta$ as the parameter scale increases, to prove the feasibility of this method in large-scale models?
4. The References section contains multiple glaring formatting errors and missing information, severely impacting the manuscript's professionalism.

**Limitations:**

The authors acknowledge computational limitations, but they evade a rigorous performance and efficiency comparison with mature BPE systems. Furthermore, there is a lack of deep discussion regarding the computational overhead of the method itself—specifically, the latency introduced by scan/sequential operations compared to standard dictionary lookup tokenization. The Impact Statement is also perfunctory and lacks depth.

**Strengths And Weaknesses:**

### Strengths ###

- The mathematical derivation of the stochastic computation graph and the application of RL variance reduction techniques are theoretically sound.
- The problem definition is clear, and the visualizations of learned token boundaries are intuitive and effective .
- Finding an adaptive tokenization scheme to replace hardcoded BPE is a critical pain point in current LLM research .
- Utilizing the output of early-exit layers as a dense baseline reward to reduce variance is a clever cross-domain synthesis .

### Weaknesses ###

- There is a "fatal evasion" in the experimental baselines. In the core evaluation (Table 1), the authors only compare against a weak Uniform baseline and other immature dynamic methods (Nawrot et al., Hwang et al.) . They completely avoid direct comparison with highly optimized traditional BPE tokenizers. Furthermore, the empirical evaluation is severely limited by the micro-scale of the 147M model; the zero-shot accuracy on downstream tasks is exceptionally low and noisy (e.g., LAMBADA at 0.086).

- The forward-pass implementation details of the sliding window scan operation could be more explicitly formalized.

- The paper champions abandoning heuristic rules for "end-to-end" learning , yet in practice compromises by enforcing a rigid "target downsample rate" constraint . This "dancing in shackles" approach undermines the claim of freely discovering language representations, significantly diminishing the practical breakthrough value of the work.

---

> ### Author Rebuttal · Authors · 2026-03-31
>
> Thank you  for the thorough review. We engage with each point below.
>
> **Target downsample rate as a heuristic.** We agree that the proposed method has *fewer* heuristics rather than *no* heuristics. However, we offer two observations. First, we are interested in performance per compute rather than raw performance. Without guidance, the model will choose to draw token boundaries at every byte (see our ablation results in response to Reviewer XMsP). The proposed loss can be viewed as a per-token reward that factors in this compute trade-off. Second, having a fixed target rate rather than a fixed per-token reward is a practical necessity: wildly varying sequence lengths across a training run would be extremely challenging for distributed training infrastructure.
>
> **Comparison with BPE.** We address this concern in detail in our response to Reviewer jqG7. In summary, we trained a BPE tokenizer with a vocabulary of 200k tokens and used the resulting boundaries to guide the autoregressive U-Net. Our method matches the performance of BPE-guided downsampling and is the only dynamic tokenization method to do so without requiring external priors.
>
> **Scale of evaluation.** Please see our response to Reviewer TUGD, where we also provide bits-per-byte measurements for the correct answers across all downstream benchmarks and models, which offer a more informative comparison than zero-shot accuracy at this scale.
>
> **Gradient variance at scale.** It is a valid observation that longer sequence lengths could cause score-function estimators to exhibit increased variance. However, this is substantially mitigated by time discounting: after 1,000 steps, the discount weight is approximately $4 \\times 10^{-5}$. We are uncertain what the most informative way to measure gradient variance across different parameter dimensions would be. We do not expect the scalar gradient of $\\pi(a)$ to vary substantially, but we welcome the reviewer to suggest a specific formula and we would be happy to provide the corresponding measurements.
>
> **Sliding window formalization.** We would first like to note that the sliding window is not necessary for the performance of our method (see our ablation results in response to Reviewer XMsP). Our current implementation offloads the tensor of shape $(w, \\text{sequence\\_length}, \\text{batch\\_size})$ to CPU and performs the computation there, incurring approximately 20\% overhead in our setup. We believe that leveraging frameworks with native scan capabilities (such as \texttt{jax.lax.scan}), advanced techniques such as speculative-decoding-style rejection sampling, or simply the pipeline parallelism inherent in distributed training setups could reduce this overhead to a negligible level. However, as we do not demonstrate this in the current work, we would recommend $w = 1$ to practitioners for now.
>
> **Reference formatting.** We thank the reviewer for pointing this out. We have normalized the conference naming and URL formatting in the revised manuscript.

---

> > ### Author Rebuttal · Reviewer_fpf9 · 2026-04-07
> >
> > My Concerns are addressed.

---

> > > ### Author Response · Authors · 2026-04-08
> > >
> > > To add to the previous discussion: this is the result for running a 147M-parameter model with $w=1$:
> > >
> > >
> > > | Model | ARC-Easy | HellaSwag | LAMBADA | PIQA | FineWeb Test |
> > > |---|---|---|---|---|---|
> > > | Ours (w=1) | **1.987 ±0.018** | **1.199 ±0.002** | **1.584 ±0.013** | **1.561 ±0.011** | **1.280 ±0.003**  |

---

### Official Review · Reviewer_XMsP · 2026-03-12

**Soundness:** 3
**Presentation:** 3
**Significance:** 3
**Originality:** 3
**Overall Recommendation:** 4
**Confidence:** 4

**Summary:**

This paper proposes an end-to-end tokenization approach that learns tokenization, or more accurately byte token merge locations, for language modeling tasks. The approach splits language model training into a small tokenization model for encoding and decoding back into bytes, which is a learned policy for tokenization that directly optimizes for the cross entropy loss of the output, while placing the bulk of the flops in the main central model. The approach is shown to work in theory, directly optimize for the cross entropy loss end-to-end,  and is empirically validated on small language model training settings versus other byte level tokenization methods, showing improved performance.

**Compliance With Llm Reviewing Policy:**

Affirmed.

**Final Justification:**

This paper proposes a good, and at a base level simple, approach to end-to-end tokenization. I like and respect the technical approach and it is clearly communicated in the paper.

My first concern addressed by the reviewers was a lack of comparisons against *any baseline* that outperformed uniform sampling. The authors responded with a baseline where the proposed method matches BPE, which lessens this concern, and I am sure that "mere engineering" could marginally outperform the BPE baseline.

In their followup reply rebuttal comment, the authors address my concern about reported downsampling rate, which was essential to know to ensure fair comparison against prior work. Now that this concern too is addressed, **I raise my score to weak accept.**

I am skeptical that performing RL on top of BPE requires a "multi-month" evaluation as the authors claim but I agree that this extra baseline may be an excessive ask. This purpose of this particular ask was for any additional baseline on top of those presented. However in a related work search I do not see significant missing public baselines, and introducing a simple approach that outperforms complex approaches such as H-Net, which the community has shown interest in, is useful for the community.

My ask to the authors is to clarify in an ablation which RL stabilization choices are essential. The experiments in the rebuttal demonstrate that these tricks didn't improve performance, however the experiments in the reply rebuttal comment say otherwise.

**Key Questions For Authors:**

How much more stable was training when implementing RL stability techniques?
Does the RL model actually encourage the model to encode every byte separately without an additional loss to push toward a target threshold?
How does performance change with respect to different target thresholds? Is there a meaningful limit to token size?
Does this approach outperform BPE with equal compute?
Do the tokenizer boundaries meaningfully continue to improve or can they be frozen midway through training?

**Limitations:**

I would like to have more discussion on the stability of this approach during training, as this could harm scaling. I believe that although the authors do not need to scale their approach for sufficient results, properly addressing potential challenges to scaling would be insightful.

**Strengths And Weaknesses:**

Soundness:
Strengths: The approach is clearly explained and well-motivated theoretically. The authors properly establish appropriate desiderata for language model tokenization and deliver on both proving that their approach optimizes toward a local minimum in this space and in practice by training models in this regime. The method is clear, well justified, technically sound, and appropriate for the task.

Weaknesses: Unfortunately, the evaluation of the method is clearly limited and leaves further analysis to be desired. The work operates at a small scale of base language models. Although it is not necessary that the authors provide experiments on larger scale models as compute can be prohibitive, without scaling as an additional dimension of results, thoroughness in base-model-sized evaluations should instead be provided. The two quantitative figures and single quantitative table do not provide enough information to validate that the approach is significantly better than others. For instance, although the trained model demonstrates lower bits-per-byte per FLOP, an established metric, this isn't validated against other metrics. Is the token-reduction rate held constant? The authors note that a lower perplexity is to be expected when fewer tokens are merged, however this is only implicitly controlled for by holding flops equal.
Furthermore, the existing baselines are concerning, as the reimplemntation of HNet, which outperforms the proposed approach in Table 1 in one benchmark, performs worse than naive Uniform boundaries. This leads to concerns about the quality of the existing baselines. Could the authors compare against BPE in the same setting? This is also comparable in a bits-per-byte evaluation per FLOP, and seems like an important baseline to be missing considering that the primary motivation for this work is to outperform BPE.
Finally, several RL techniques for denoising the gradient are introduced, such as denoising by using the distributional output of the tokenizer model. However, there are no ablations to suggest that these additional techniques are necessary. Including these would help readers understand this work.

Presentation: Strengths: This paper is very well written and understandable. The narrative is easy to follow and the paper is well contextualized with respect to prior work. The motivation makes sense.
Weaknesses: Please modify the color scheme of the qualitative figures (e.g Figure 2) to support accessible color schemes, such as the commonly used Plasma. The output probability of token boundary selection looks more binary than necessary. Plots in Figures 3,4 are also small and could use error bars to contextualize performance.

Significance: Improving tokenization to be an end-to-end optimization objective is a useful and important problem. In particular, the end-to-end RL approach which seeks to minimize loss is easy to understand and well motivated, and the fact that this works is notable compared to all existing methods that use trained models, which use heuristics.
The improvements are empirically modest, however this technique is a simplification of additional approaches in that it proposes the minimally complex approach to meet the stated desiderata, which are well motivated. Demonstrating that the simplest possible theoretically motivated approach for tokenization works is a useful contribution to the community.

Originality: This paper is largely applications focused in establishing any system that can train a tokenizer end-to-end. The existing RL techniques are well established, however they are executed well in the paper, and RL with straight through estimation has not yet been applied to this task.

I am recommending a weak reject due to the lack of experiments to convince me that the existing approach outperforms or matches comparable baselines, and due to a lack of ablations that demonstrate how implemented features for RL stability behave.

---

> ### Author Rebuttal · Authors · 2026-03-31
>
> ## Response to Reviewer XMsP
>
> Thank you for the detailed and constructive review. We address each point below.
>
> **Accessible color schemes and figure formatting.** We thank the reviewer for this suggestion. We have updated the color scheme of the qualitative figures (e.g., Figure 2) to use accessible palettes and have increased the size of Figures 3 and 4 with added error bars in the revised manuscript.
>
> **Binary appearance of boundary probabilities.** This is expected behavior: by the end of training, the model converges to a highly consistent token boundary estimate, resulting in near-binary boundary probabilities.
>
> **Concern about baseline quality (HNet).** Our implementation of HNet was performed by directly copying the authors' code for the downsampler and upsampler, and was rigorously tested. We provide our code so that this can be validated by others. One potential explanation for the unexpectedly weak performance is that the original authors have since stated that their method did not perform well at small scale (https://x.com/_albertgu/status/1965853622819787232). Unfortunately, we do not have the compute budget to verify this at larger scales.
>
> **RL stability techniques and ablations.** We arrived at our method by qualitatively analyzing training dynamics in the first $<10\\%$ of training on small models. Without any stabilization techniques, we observed the models to collapse into trivial solutions.
>
> To ablate these design choices, we trained 40M-parameter models (with 4 byte-level, 6 token-level, and 4 byte-level layers) on 1.5B bytes of FineWeb using the same setup described in Section 4. We ablated the following components: no time discounting ($\\gamma = 1$), no batch-relative advantages, no early exit-relative rewards, no sliding window ($w = 1$) and no consistency loss ($\\lambda_{\\text{target}} = 0$). Overall, we observe that removing the consistency loss causes the model to quickly converge to a tokenize-every-byte solution. Batch-relative advantages and the sliding window can be dropped without significantly affecting the resulting tokenization strategy, whereas removing the early exit or time discounting diminishes, though does not completely negate, the frequency with which the model tokenizes whitespace.
> We report results in the table below. At this scale, there does not appear to be a significant effect on downstream loss from small differences in tokenization strategy. We have added these results, along with corresponding heatmaps, to the appendix of our revised manuscript.
>
> | Ablation | Downsample Rate at convergence | FineWeb Validation BPB | Fraction of Whitespace with token boundaries |
> |---|---|---|---|
> | Baseline | 0.196 | 1.548 | 0.99 |
> | no time discounting ($\gamma=1$) | 0.215 | 1.542 | 0.41 |
> | no batch-relative advantages | 0.208 | 1.543 | 0.99 |
> | no early-exit relative rewards | 0.200 | 1.562 | 0.72 |
> | no sliding window ($w=1$) | 0.196 | 1.511 | 0.99 |
> | no consistency loss ($\lambda_{target} = 0$) | 1.0 | 1.394 | 1.0 |
>
> **Comparison with BPE.** Our method matches BPE-guided downsampling. Please see our response to Reviewer jqG7 for the full experimental details and results.
>
> **Different target thresholds.** For our small-scale experiments, we observed that smaller tokens (i.e., less aggressive downsampling) tend to perform better. However, scaling the downsampling rate beyond what is achievable with pure BPE still produced meaningful token boundaries.
>
> **Freezing boundaries mid-training.** Anecdotally, we observe that boundaries stabilize after approximately 10\% of the training run, suggesting that freezing them is a viable option. We have added plots of the selected action cross entropy, $-\\log \\pi(a)$, over the course of training to demonstrate this convergence behavior.

---

> > ### Author Rebuttal · Reviewer_XMsP · 2026-04-04
> >
> > **Comparison with BPE**
> > In the response I am referred to, the rebuttal quotes: "Dynamic tokenization methods have applications that BPE inherently cannot address, such as continuous domains (e.g., audio or video) or the downsampling of BPE tokens themselves." Simultaneously, I am told that H-Net collapses at this compute scale. Comparing against another method in a regime where it collapses, as well as not comparing against a simpler non-collapsing method in the primary experiments, raises concerns for the performance of this method. If performance is to not exceed BPE tokenization, why should we use a dynamic tokenizer at all? Why not perform RL to further compress BPE tokens to attempt to outperform related work. I sympathize with being unable to run experiments at a greater compute scale, however this paper does not provide a reason to suspect that the proposed approach would be significantly better at a larger scale (such as showing a scaling law curve instead of a single model size). This is not strong enough evaluation.
> >
> > > At this scale, there does not appear to be a significant effect on downstream loss from small differences in tokenization strategy.
> >
> > Then why are there significant sections devoted to these components? It seems like nearly every component described on Page 4 doesn't actually affect performance. The evaluation in an ICML submission should be significantly more scientific and reproducable than the standard of "We arrived at our method by qualitatively analyzing training dynamics in the first $<10\%$ of training on small models."
> >
> > **Different target thresholds**
> >
> > * For the "no consistency loss ($\lambda_{target} = 0$)" in the Ablation, I'd recommend comparing in a compute-held-equal regime, in which case the BPB may be worse.
> > * It is shown here in the rebuttal table that (as I mentioned in my original rebuttal) performance is better with a greater downsample rate on the tokenizer. However, downsample rates affect necessary compute. How does the downsample rate of the provided method compare against the baselines? Without this value, which I asked for in my original rebuttal, performance improvements from the proposed method may instead be from having an inconsistent downsample rate with the baselines. As a result, I can not be confident in the proposed method without this number and I maintain weak reject.

---

> > > ### Author Response · Authors · 2026-04-08
> > >
> > > **Downsampling rates across baselines.** We apologize for the oversight in not reporting this earlier. Although training FLOPs are held constant across all methods, the reviewer is correct that inference FLOPs—which depend on the downsampling rate—will also affect benchmark scores. The exact downsampling rates at convergence on FineWeb for the different methods are as follows:
> > >
> > > | Model | Downsampling rate |
> > > |---|---|
> > > | H-Net (Hwang et al.) | 0.206 |
> > > | Dynamic (Nawrot et al.) | 0.200 |
> > > | Uniform | 0.200 |
> > > | Ours |  0.204 |
> > > | Ours ($\gamma=1$) | 0.202 |
> > > | BPE guidance | 0.207 |
> > >
> > > Notably, our method achieves a slightly *lower* downsampling rate than both BPE Guidance and H-Net, meaning that we are giving a mild advantage to those baselines rather than benefiting from an inconsistent comparison.
> > >
> > > **Effect of variance reduction components on performance.** We respectfully disagree with the characterization that nearly every component on Page 4 does not affect performance. To demonstrate this, we trained a larger 147M-parameter model using the same configuration as the other 147M runs but with no time discounting ($\\gamma = 1$) and observe the following scores:
> > >
> > > | Model | ARC-Easy | HellaSwag | LAMBADA | PIQA | FineWeb Test |
> > > |---|---|---|---|---|---|
> > > | Ours ($\\gamma=1$) | 2.021 ±0.018 | 1.281 ±0.002 | 1.708 ±0.011 | 1.640 ±0.011 | 1.343 ± 0.003|
> > >
> > > These results show a meaningful degradation relative to our full method, confirming that the variance reduction techniques do contribute to final performance. More broadly, we believe that when presenting an existence proof to the research community—as we do in this paper—it is prudent to err on the side of including stabilization techniques that we have reason to believe are helpful, even if not every component produces a large isolated effect at small scale.
> > >
> > > **Reproducibility of method development.** We agree that the evaluation in an ICML submission should be scientific and reproducible, which is why we did not cite our qualitative exploratory analysis as evidence anywhere in the manuscript. Qualitative analysis of training dynamics is a valid and common approach to method development under compute constraints, and we do not believe we should be penalized for transparency about our development process. Building an entire experimental setup before conducting any exploratory work would tend toward incrementalism. We would like to highlight the reviewer's own observation that "demonstrating that the simplest possible theoretically motivated approach for tokenization works is a useful contribution to the community." We agree that our paper is primarily an existence proof, and while we have not been able to ablate every component at every scale, we believe the evidence presented is sufficient to support our claims.
> > >
> > > **Why not perform RL to further compress BPE tokens.** What the reviewer is suggesting constitutes a multi-month, compute-intensive research project requiring new infrastructure, evaluation pipelines, and extensive experimentation—a project that could be undertaken by anyone who reads our paper. We believe it would be strictly worse for the research community if only we were to work on this direction while keeping the foundational method unpublished. We highlight this to emphasize that our paper satisfies the standard of providing "a contribution that others are likely to build on," even if BPE-based tokenization produces similar results at the byte level in our current experiments.

---

### Official Review · Reviewer_jqG7 · 2026-03-12

**Soundness:** 3
**Presentation:** 3
**Significance:** 3
**Originality:** 4
**Overall Recommendation:** 4
**Confidence:** 2

**Summary:**

The paper proposes a method to learn token boundaries during training by moving tokenization inside LLM architectures, as an alternative to algorithms such as BPE that produce fixed boundaries before training. The authors show that this is indeed possible by means of reinforcement learning techniques such as score function estimation, paired with strategies such as time discounting to reduce the estimator's variance. The authors show that their method outperforms prior state-of-the-art boundary-learning techniques such as straight-through estimators.

**Compliance With Llm Reviewing Policy:**

Affirmed.

**Final Justification:**

The paper studies a problem that is very novel and very relevant. While I agree with other reviewers that the experiments provided are limited, I still think that the overall technical contribution of the work is valuable.

**Key Questions For Authors:**

I have the following questions for the authors:
1. For a non-expert like me, it would be nice to see how your method compares to standard fixed tokenizers like BPE, to better understand how far we still are from state-of-the-art solutions.
2. It would also be nice to see how your method performs on bigger models and datasets. As you said, prior work shows promising results in moving tokenization inside the LLM architecture at scale. Yet no results of this kind are shown for your method.
3. It is not clear to me why you claim (Line 68) that U-Net-like architectures are necessary to satisfy the desiderata in Sec. 2.1. Could you elaborate more on that point?

I would also ask the authors to fix the following typos:
- The paragraph starting around Line 102 second column is syntactically wrong;
- The same holds for the paragraph starting around Line 215 first column;
- The first reference in the bibliography in Page 9 needs to be fixed.

**Limitations:**

Yes.

**Strengths And Weaknesses:**

**Strengths**

As a non-expert on tokenization, I think that the idea of learning token boundaries during training is very interesting and deserves further study. The authors' explanation of the topic is very clear. For example, I like the first-principles approach that the authors take qwhen they state in Sec. 2.1 the desiderata that a boundary-learning method has to satisfy to be practical. The idea of basing their method on reinforcement learning and in particular on score estimation is very reasonable. The sanity-check qualitative results in Figure 2 confirm the soundness of their methodology.

**Weaknesses**

On the downside, the paper feels more like a first step in a new direction than a full-fledged study. The numerical experiments are limited and carried out on very small architectures. Qualitative results like those in Figure 2 are useful as sanity-checks on the learned boundaries. However, the boundaries learned still seem quite trivial. I imagine that the final performance to still be far from that obtained with BPE, even though the authors only show comparison with other learned tokenizers and not with fixed ones like BPE.

---

> ### Author Rebuttal · Authors · 2026-03-31
>
> Thank you the thoughtful and constructive feedback. We address each point below.
>
> **Comparison with BPE (Question 1).**
>
> We agree that understanding the gap relative to BPE is a valid and informative line of inquiry. To address this, we trained a BPE tokenizer with a vocabulary size of 200k tokens, achieving a downsampling rate of 0.207 on the FineWeb training set, and used the resulting boundaries to guide tokenization in the autoregressive U-Net. An important caveat is that BPE leaks information forward: for example, "tokens" is encoded as a single token, whereas "tokenization" is split into "token" and "ization." Thus, if one sets $a_i = 1$ at the byte "n," the model already knows that the next byte will not be "s." To rectify this information leakage, we apply a single right shift to the boundary indicators. For instance, the sentence "hello world tokenization" receives boundary values:
>
> $$x_i: \\text{h e l l o \\_ w o r l d \\_ t o k e n i z a t i o n}$$
> $$a_i: \\text{0 0 0 0 0 1 0 0 0 0 0 1 0 0 0 0 0 1 0 0 0 0 0 0}$$
>
> We trained a 147M-parameter model using this scheme with the same hyperparameters described in Section 4. This achieves  an almost identical FineWeb test BPB to our method ($1.2990 \pm 0.0031$); see the table given to reviewer TUGD for the full output. Notably, our method is the only dynamic tokenization method to recover the performance of BPE-guided downsampling without requiring these external priors.
>
> We appreciate all reviewer's interest in a BPE baseline. We wish to clarify that the primary purpose of our paper is to compare dynamic tokenization methods on autoregressive U-Nets using identical model hyperparameters. Dynamic tokenization methods have applications that BPE inherently cannot address, such as continuous domains (e.g., audio or video) or the downsampling of BPE tokens themselves. We study byte-level language modeling because it is a relatively compute-cheap regime where we already know what good tokenization *should* look like. We have refined the arguments in our manuscript to reflect this motivation.
>
> **Scaling to larger models (Question 2).** We agree that evaluating on larger models and datasets would strengthen the paper. Unfortunately, our compute budget is limited and does not permit such experiments at this time.
>
> **Necessity of U-Net-like architectures (Question 3).** We restrict ourselves to models consisting of a series of sequence-to-sequence layers operating on either bytes or tokens. Such a model can always be formulated as a sequence of upsampling/downsampling layers interleaved with the sequence-to-sequence layers. For autoregressive models, a given token can only depend on prior bytes and not on future ones (unlike, for example, in MaNТА), which means that the token boundaries must be discrete in nature—hence $a_i$ are samples rather than continuous values. Furthermore, in order to minimize non-coalesced memory accesses, we wish to minimize the number of upsampling/downsampling layers. While this is not a hard constraint, we study the simple case of a single downsampling step followed by a single upsampling step. We believe that having multiple layers of tokenization or multiple different tokenizations per sequence could be fruitful avenues for future work. We have refined the arguments in our manuscript to make this reasoning more precise.
>
> **Typos.** We thank the reviewer for identifying these issues and have corrected them in the revised manuscript.

---

> > ### Author Rebuttal · Reviewer_jqG7 · 2026-04-02
> >
> > I thank the authors for the additional experiments on BPE. While experiments larger models and datasets are still missing, I understand that it would require significant compute budget. Even without these experiments, I appreciate that the paper provides a nice solution to a complex problem.

---

### Decision · Program_Chairs · 2026-04-30

**Decision:**

Accept (regular)

**Comment:**

This paper proposes to replace hardcoded tokenization such as BPE to an end-to-end learned process. By formulating token boundary selection as a RL problem and applying variance reduction techniques, the authors successfully demonstrate a viable path toward fully differentiable language modeling.

The idea of using score function estimation to learn discrete token boundaries is novel, going beyond the limitations of straight-through estimators. The application of RL stability techniques (discounting, early-exit rewards) effectively prevents trivial solution collapse.

During rebuttal, the authors provided a critical BPE-guided baseline, showing that the proposed method matchs the performance of BPE-guided downsampling without requiring external priors. The authors also added bits-per-byte (BPB) evaluations and ablations, showing that the variance reduction components is meaningful to the final performance at some scale.

Several reviewers noted that the evaluations were conducted at a relatively small scale (147M parameters), they reached a consensus that the work serves as a valuable existence proof for end-to-end tokenization. It is likely that the paper provides a foundational contribution that the community can build upon.